# PDUNet: Proximal-Guided Deep Unrolling Network for Time Series Forecasting

## Abstract

Recent time series forecasting methods have increasingly incorporated model-driven formulations and data-driven inference to leverage their complementary strengths in interpretability and pattern learning, achieving remarkable results in many practical tasks. However, in most existing methods, the forecasting target is only used as a training supervision signal rather than being explicitly incorporated into the modeling process, making it difficult for models to leverage future modality information for structural constraint and path guidance. As a result, the modeling structure and inference path remain isolated, lacking a clear optimization objective, and ultimately degenerate into an uncontrollable and uninterpretable static mapping process. To address these challenges, we propose Proximal Deep Unrolling Network (PDUNet), a coupled and closed-loop forecasting framework that starts from modeling, leverages data-driven mechanisms for solution, and progressively optimizes the inference path. Specifically, we formulate an optimizable forecasting equation based on the coupling between future variables and historical inputs, and adopt a proximal optimization algorithm to perform step-wise decoupling and update. This optimization process is further unfolded into a dual-branch state space structure, where the Temporal-SSD captures temporal evolution through causal modeling, while the Channel-SSD employs a non-causal mechanism to model global interactions among variables, jointly enabling progressive inference and dynamic prediction. Experiments on eight public benchmark datasets show that PDUNet outperforms existing state-of-the-art models in long-term forecasting tasks.

## 1 Introduction

Forecasting has always played a crucial role in the process of human exploration of the unknown. Particularly, time series forecasting has demonstrated extensive value in a wide range of fields, such as energy scheduling Wang et al. (2019) Joshua et al. (2024), traffic management Yu et al. (2017) Hightower et al. (2024), financial modelling Tang et al. (2022) Sako et al. (2022) and weather forecasting Karevan & Suykens (2020) Zhang et al. (2024), and has become one of the core technologies that drive intelligent decision making.

In time series forecasting, model-driven and data-driven methods play essential and complementary roles. Model-driven approaches typically rely on analytical formulations, dynamical priors, or statistical assumptions to explicitly construct the forecasting objective, which improves interpretability and controllability. In contrast, data-driven methods utilize the high capacity of deep neural networks to capture nonlinear dependencies and long-range temporal correlations, enhancing the ability to represent complex temporal patterns. Recent studies have increasingly explored combining these two paradigms by using model-based logic to define inference structures, and employing data-based optimization to express and refine the prediction trajectory. For example, Koopman theory achieves characterisation of complex dynamics through dimensionality-increasing linear modelling Liu et al. (2023b), while the Score Matching method combines diffusion modelling with the solution distribution of physical constraint learning inverse problems Holzschuh et al. (2023). The other class of methods focuses on statistical features such as trends and periodicity in signals, combining sequence decomposition and deep optimisation to enhance modeling performance. For example, FEDformer Zhou et al. (2022) models long-range dependencies based on frequency-domain sparsity,

while ETSformer Woo et al. (2022) explicitly models trend and seasonal components by integrating exponential smoothing and Fourier transforms.

Although the above approaches have established preliminary connections between structural modeling and expressive capacity, and have achieved significant progress on various forecasting tasks, several fundamental limitations remain. Most existing methods treat the future sequence solely as a supervision signal during training, without explicitly incorporating the prediction target into the modeling process. This prevents the model from leveraging information from the future modality for constraint and guidance. Consequently, model structures are often heuristically designed, while the inference process is independently learned through black-box networks, lacking structural consistency and semantic alignment with the forecasting objective. Furthermore, this disjoint design causes the prediction procedure to degenerate into an uncontrollable and non-interpretable static mapping, making it difficult to support progressive convergence or feedback correction through intermediate states. As illustrated in Figure 1(a), mainstream paradigms typically rely on direct generation from historical sequences, rather than embedding the future variables as optimizable targets within a dynamically evolving inference trajectory.

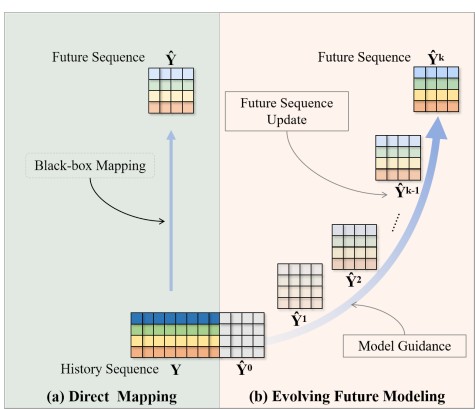

Figure 1: (a) In the mainstream prediction paradigm, models generate future results statically based on historical sequences; (b) In the PDUNet concept, the prediction target is modelled as an optimisable variable that is gradually updated under structural guidance to form a dynamically evolving modelling path.

To address the limitations of static forecasting formulations and enable a controllable, path-aware prediction process, we propose a novel framework called Proximal-Guided Deep Unrolling Network (PDUNet), which integrates inductive priors with data-driven modeling through an explicit optimization formulation. As illustrated in Figure 1(b), PDUNet formulates time series forecasting as a step-wise optimization problem, where the future sequence is treated as a latent variable and iteratively inferred through structured updates. Specifically, we introduce a model-driven forecasting equation based on proximal operators, which allows future values to be explicitly included as optimization targets under a well-defined objective. As shown in Figure 2, this optimization problem is then unfolded into a deep neural architecture, where each stage mimics one iteration of the solution process. The proximal operator provides interpretable update rules rooted in modeling assumptions, while the unrolling process parameterizes the reasoning path with learnable modules. This design integrates model-driven formulation and data-driven inference into a unified, controllable framework that enables progressive refinement of predictions and improves long-term forecasting performance.

Our contributions can be summarised as follows:

- We develop a novel forecasting framework where model-based mechanisms are tightly integrated with distribution priors through iterative updates and stepwise feedback, enabling a dynamic and unified modeling process.

- PDUNet formulates time series forecasting as a global optimization problem, in which future variables are explicitly modeled as optimization targets and iteratively solved under proximal guidance, enabling a controllable and interpretable prediction trajectory.

- We design a Dual-SSD module that captures structural dependencies along both temporal and channel dimensions, which enhances the ability of the model to perceive multiscale dynamics in high-dimensional time series.

- We conducted extensive experiments on multiple datasets. The results show that PDUNet outperforms state-of-the-art baselines.

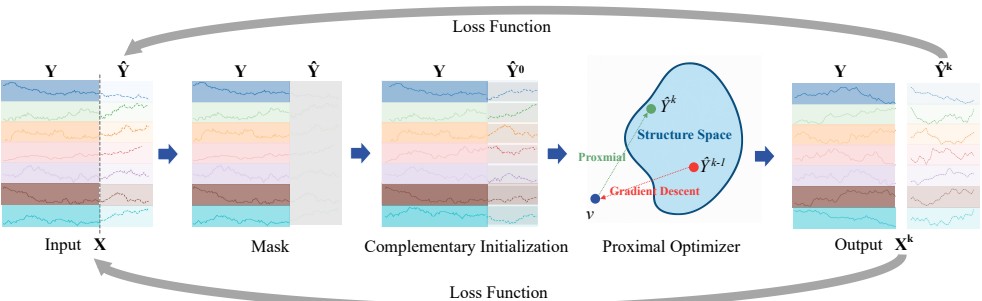

Figure 2: Schematic diagram of the predictive modelling and optimisation process for PDUNet. The model first receives the input sequence $\mathbf{X}$ and divides it into the historical part $\mathbf{Y}$ and the part to be predicted $\hat{\mathbf{Y}}$. In the structural modelling space, the PDU formalises $\hat{\mathbf{Y}}$ as variable optimisation paths and iteratively updates them with gradient descent and proximal projection at each stage.

## 2 PRELIMINARY STUDIES

We formulate forecasting as a structured optimization problem over the future variable $\hat{\mathbf{Y}}$. A unified representation is defined by concatenating historical observations $\mathbf{Y}$ and the prediction target $\hat{\mathbf{Y}}$. An observation-guided objective is optimized via proximal gradient descent. This yields a structured and iterative prediction process.

### 2.1 PROBLEM FORMULATION

In time series modeling, we denote the input sequence $\mathbf{X} \in \mathbb{R}^{N \times T}$ as the concatenation of history $\mathbf{Y} \in \mathbb{R}^{N \times R}$ and future $\hat{\mathbf{Y}} \in \mathbb{R}^{N \times L}$, where $T = R + L$. The prediction task is to estimate $\hat{\mathbf{Y}}$ from $\mathbf{Y}$.

To incorporate prior structure, we introduce a selection matrix $\mathbf{E}$ that extracts the historical part from $\mathbf{X}$, leading to the reconstruction objective

$$\min_{\mathbf{X}} \ \tfrac{1}{2}\|\mathbf{Y} - \mathbf{X}\mathbf{E}\|_F^2 + \lambda \, \mathcal{J}(\mathbf{X}), \tag{1}$$

where $\mathcal{J}(\cdot)$ denotes structural regularization. Extending this to future completion, we model

$$\mathbf{X} = \mathbf{Y}\mathbf{A} + \hat{\mathbf{Y}}\mathbf{B} + \mathbf{N}, \tag{2}$$

and formulate the prediction as

$$\min_{\hat{\mathbf{Y}}} \ \tfrac{1}{2}\|(\mathbf{Y}\mathbf{A} + \hat{\mathbf{Y}}\mathbf{B})\mathbf{E} - \mathbf{Y}\|_F^2 + \lambda \, \mathcal{J}(\hat{\mathbf{Y}}), \tag{3}$$

with $\mathbf{A}, \mathbf{B}$ as learnable parameters. This transforms forecasting into an optimization over $\hat{\mathbf{Y}}$, ensuring interpretable and controllable modeling of historical–future dependencies.

### 2.2 OPTIMIZATION OBJECTIVE

Based on the structural prior in Eq. 3, we formulate the prediction task as an optimization problem over the future variable $\hat{\mathbf{Y}}$:

$$\min_{\hat{\mathbf{Y}}} \ \frac{1}{2} \left\|(\mathbf{Y}\mathbf{A} + \hat{\mathbf{Y}}\mathbf{B})\mathbf{E} - \mathbf{Y}\right\|_F^2 + \lambda \, \mathcal{J}(\hat{\mathbf{Y}}), \tag{4}$$

where $\lambda$ is a hyperparameter and $\mathcal{J}(\cdot)$ denotes the structural regularization term. This formulation treats the predicted future $\hat{\mathbf{Y}}$ as an optimization target constrained by a model-based prior, enabling an interpretable and controllable prediction mechanism.

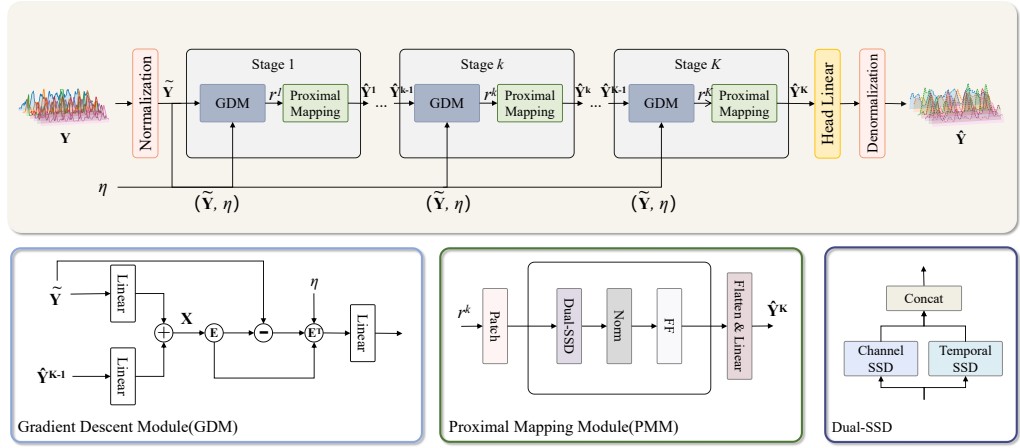

Figure 3: The overall architecture of PDUNet. The architecture structures the iterative optimisation process into a learnable network, comprising multiple stages of gradient descent modules (GDM) and proximal mapping modules (PMM), which alternately perform gradient updates and prior guidance.

### 2.3 GRADIENT DERIVATION AND PROXIMAL UPDATE

In order to efficiently solve the above objective function (eq. 4), we use the proximity gradient method to iteratively expand it. First, we take $g(\hat{\mathbf{Y}})$:

$$g(\hat{\mathbf{Y}}) = \frac{1}{2} \left\| (\mathbf{Y}\mathbf{A} + \hat{\mathbf{Y}}\mathbf{B})\mathbf{E} - \mathbf{Y} \right\|_F^2. \tag{5}$$

To efficiently solve this problem, we construct a proximal quadratic approximation of the objective around the current iterate $\hat{\mathbf{Y}}^{(\mathbf{k})}$:

$$Q(\hat{\mathbf{Y}}, \hat{\mathbf{Y}}^{(k)}) = g(\hat{\mathbf{Y}}^{(k)}) + \left\langle \hat{\mathbf{Y}} - \hat{\mathbf{Y}}^{(k)}, \nabla g(\hat{\mathbf{Y}}^{(k)}) \right\rangle + \frac{1}{2\eta} \left\| \hat{\mathbf{Y}} - \hat{\mathbf{Y}}^{(k)} \right\|_F^2 + \lambda_f f(\hat{\mathbf{Y}}), \tag{6}$$

where $\eta \in \mathbb{R}$ is the step size, $\lambda_f f(\hat{\mathbf{Y}})$ denotes the regularization term, and $\nabla g(\hat{\mathbf{Y}}^{(k)})$ is the gradient of $g(\cdot)$ evaluated at the $k$-th iterate $\hat{\mathbf{Y}}^{(k)}$.

By minimizing $Q(\hat{\mathbf{Y}}, \hat{\mathbf{Y}}^{(k)})$, we obtain the proximal gradient update:

$$\hat{\mathbf{Y}}^{(k+1)} = \text{prox}_{\eta, \lambda \mathcal{J}} \left( \hat{\mathbf{Y}}^{(k)} - \eta \cdot \nabla g(\hat{\mathbf{Y}}^{(k)}) \right), \tag{7}$$

where $\text{prox}_{\lambda_f \eta}(\cdot)$ is the proximal operator that enforces the prior regularization $f(\cdot)$.

The final gradient unrolling and proximal update processes can be jointly expressed as:

$$\hat{\mathbf{Y}}^{(k+1)} = \text{prox}_{\eta, \lambda \mathcal{J}} \left( \hat{\mathbf{Y}}^{(k)} - \eta \left( (\mathbf{Y}\mathbf{A} + \hat{\mathbf{Y}}^{(k)}\mathbf{B})\mathbf{E} - \mathbf{Y} \right) \cdot (\mathbf{B}\mathbf{E})^\top \right), \tag{8}$$

where $\eta$ is the step size hyperparameter, $\mathcal{J}(\cdot)$ denotes the prior regularisation term, and $\text{prox}_{\eta, \lambda \mathcal{J}}(\cdot)$ denotes the weighted regularised proximal mapping.

To better illustrate this iterative optimisation process, we use an approximate expansion method to summarise the entire prediction process as the algorithm 8 in the appendix. The algorithm takes historical observations and a structural projection matrix as input, and iteratively refines the prediction through gradient descent and proximal updates.

## 3 PDUNET FRAMEWORK

The PDUNet framework we propose is shown in Figure 3. First, initial prediction variables are constructed via complementary initialization. Secondly, the gradient descent module (GDM) and the

proximal mapping module (PMM) iteratively update the predictions. Finally, the refined predictions are output as future results that follow dynamic patterns. To enhance structural perception, a dual-path feature modeling mechanism is incorporated in PMM to extract temporal and channel-specific representations. This section introduces the framework in three parts: first, the evolution of the state modeling module; secondly, the step-by-step prediction updating mechanism; and finally, the training objective and loss function.

## 3.1 State Modelling Modules

To effectively model the dynamic dependencies in multivariate time series, we adopt the State Space Duality (SSD) framework Dao & Gu (2024) as the foundation of our modelling design. Building on this foundation, we employ its improved modules, namely the Hidden State Mixer-Based SSD (HSM-SSD) and the Causal Hidden State Mixer-Based SSD (CHSM-SSD), to progressively enhance computational efficiency and modelling capacity. The standard SSD and its non-causal extension, NC-SSD, serve as essential components in the overall design. Due to space limitations, the specific formulas and derivations are provided in Appendix.

### 3.1.1 Hidden State Mixer-Based SSD (HSM-SSD).

To further reduce computational complexity, an effective mechanism based on hidden state mixers (HSM-SSD) is proposed for SSDs. This mechanism migrates all high-cost operations to a shared compressed state space, thereby compressing the nonlinear computational burden of token dimensions while maintaining modelling capabilities. The mechanism first constructs a shared compressed state based on the input sequence, and then completes gate fusion and output generation in the state space.

Let the input sequence $\mathbf{x}_{\text{in}} \in \mathbb{R}^{L \times D}$, introduce a set of importance weights $\boldsymbol{a} \in \mathbb{R}^L$, a state-channel mapping matrix $\mathbf{B} \in \mathbb{R}^{L \times N}$ and the normalisation vector $\mathbf{1}_N \in \mathbb{R}^N$. The input is first mapped to a low-dimensional space:

$$\mathbf{x}_{\text{proj}} = \mathbf{x}_{\text{in}} \mathbf{W}_{\text{in}}, \quad \mathbf{W}_{\text{in}} \in \mathbb{R}^{D \times D}. \tag{9}$$

Next, compressed states are generated based on a weighted discretisation mechanism:

$$\mathbf{h}_{\text{in}} = \left( \mathbf{a}^\top \mathbf{1}_N \odot \mathbf{B} \right)^\top \mathbf{x}_{\text{in}}, \quad \mathbf{h} = \mathbf{h}_{\text{in}} \mathbf{W}_{\text{in}}, \tag{10}$$

where $\odot$ denotes element-by-element multiplication and $\mathbf{W}_{\text{in}}$ is the state projection parameter. This step achieves a compressed projection of the input sequence, and the computational complexity is reduced from $\mathcal{O}(LD^2)$ to $\mathcal{O}(ND^2)$, which significantly relieves the computational burden under long sequences.

The nonlinear computation is then fully migrated to be performed in state space to avoid repetitive operations in the original token space. Channel mixing and gating operations are performed on the state representation $\mathbf{h}_{\text{in}}$ to generate the updated state output:

$$\mathbf{x}_{\text{out}} = f(\mathbf{y}) = \text{Linear}\left(\mathbf{y} \odot \sigma(\mathbf{z})\right) = \left(\mathbf{Ch} \odot \sigma(\mathbf{x}_{\text{in}} \mathbf{W}_z)\right) \mathbf{W}_{\text{out}} \approx \mathbf{C}\left(\left(\mathbf{h} \odot \sigma(\mathbf{h}_{\text{in}} \mathbf{W}_z)\right) \mathbf{W}_{\text{out}}\right), \tag{11}$$

where $\mathbf{W}_z, \mathbf{W}_{\text{out}} \in \mathbb{R}^{D \times D}$ are the parameters of the gating and output mapping, and $\mathbf{C} \in \mathbb{R}^{L \times N}$ is the decoding matrix of the state to token. This design allows the mixing, gating and output mapping operations to be performed in the state space, and the final reconstruction of the original token space output is done by $\mathbf{C}$ linearly, thus effectively reducing the overall computational complexity. The overall scheme reduces the computational complexity of the output stage while maintaining the modelling capability, achieving significant computational speedup, which is particularly suitable for large-scale multivariate modelling tasks.

The structure of HSM-SSD, as illustrated in the upper part of Figure 4, can be summarized in four main stages. First, the input token $\mathbf{x}$in is projected into several intermediate representations via a shared linear transformation, yielding the state transition parameters, decoding matrix, gating vector, and other auxiliary components. Next, the continuous transition parameters are discretized using a structured operator to obtain the hidden state dynamics. The state input $\mathbf{h}$in is then computed by applying a linear combination over token embeddings, followed by a gating mechanism and output projection to produce the final representation $\mathbf{x}_{\text{out}}$. This entire pipeline enables efficient token-to-state transformation, gated state update, and state-to-token decoding, while maintaining low computational complexity.

### 3.1.2 CAUSAL HIDDEN STATE MIXER-BASED SSD (CHSM-SSD).

Based on HSM-SSD, to introduce causality constraints, CHSM-SSD replaces the attentional weights $\boldsymbol{a} \in \mathbb{R}^L$ in the state construction with a lower triangular mask matrix $\mathbf{A} \in \mathbb{R}^{L \times L}$, thus ensuring that each state depends only on current and past tokens. Correspondingly, the state construction formula is updated to:

$$\boldsymbol{h}_{\text{in}} = (\mathbf{A} \odot \mathbf{B})^\top \boldsymbol{x}_{\text{in}}. \tag{12}$$

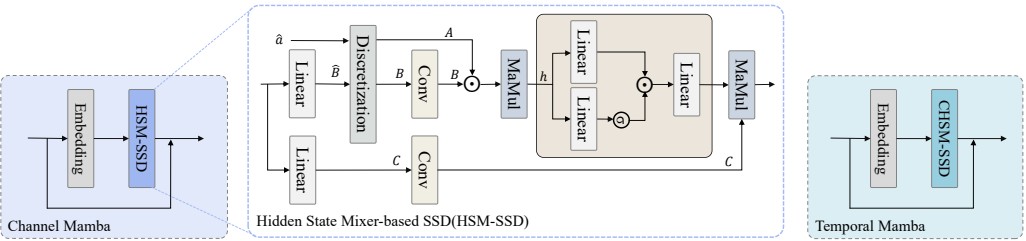

Figure 4: Composition of channel mamba and time mamba. where CHSM-SSD has the same structure as HSM-SSD, with the only difference that the weight vector $a$ is replaced by a lower triangular matrix $A$ to generate causal relationships.

## 3.2 PROXIMAL UNROLLING ARCHITECTURE

In this section, we introduce the iterative prediction process of PDUNet, which models future sequence prediction as an optimisable variable and uses alternating iterations of GDM and PMM to gradually approximate the target sequence. This mechanism supports end-to-end training and progressive optimisation, and has good convergence and interpretability.

### 3.2.1 GRADIENT DESCENT MODULE (GDM).

The module implements a structured modelling of the gradient direction in the optimisation objective, corresponding to the gradient expansion term in Equation 8. Specifically, we use two linear transformation modules to simulate matrix multipliers $\mathbf{A}$ and $\mathbf{B}$, respectively, to implement the feedback of future sequences on the reconstruction residuals. The entire process is shown in the lower left corner of Figure 3, the current predicted variable $\hat{\mathbf{Y}}^{(k)}$ and historical information $\mathbf{Y}$ are mapped linearly and interact with the residual matrix $\mathbf{X}$. The update magnitude is controlled by the step size $\eta$, thereby generating a structure-aware gradient update direction, which serves as input for the proximal mapping module in the next stage.

The output of GDM is an intermediate variable $\hat{\mathbf{Y}}^{(k)} \in \mathbb{R}^{N \times r}$ in a low-rank structure space, where $N$ is the number of channels and $r$ denotes the feature length after compression modelling, which is used to explicitly reduce the dimension of the optimisation variables. In Figure 3, we explicitly denote the output of this low-rank process as $r$ to facilitate its subsequent stepwise updating and decoding by the proximal mapping module (PMM).

### 3.2.2 PROXIMAL MAPPING MODULE (PMM).

This module performs structured modelling and nonlinear transformation on the low-rank residual features $\mathbf{r}^{(k)} \in \mathbb{R}^{N \times r}$ output by GDM to generate the prediction variables $\hat{\mathbf{Y}}^{(k)}$ for the current stage. As shown in Figure 3, the PMM module first performs patch segmentation and embedding on $\mathbf{r}^{(k)}$ to construct a local time block representation. Then, the input is modelled by the Dual-SSD module to capture the dual-axis structure of time and channel, generating a joint representation $\mathbf{Z} \in \mathbb{R}^{N \times r}$. This is further transformed through normalisation and a feedforward network to produce the stage output:

$$\hat{\mathbf{Y}}^{(k)} = \text{FFN}(\text{LayerNorm}(\mathbf{Z})) \in \mathbb{R}^{N \times r}. \tag{13}$$

Among them, Dual-SSD consists of Channel SSD and Temporal SSD connected in parallel, as shown in Figure 4, which perform structural perception processing along the channel axis and time axis, respectively, and finally perform parallel transformation on the merged results.

## 3.3 LOSS FUNCTION

In order to balance prediction accuracy and optimisation stability, we designed a total loss function consisting of the main task loss and stage regularisation terms.

The main loss measures the difference between the predicted sequence $\hat{\mathbf{Y}}$ and the true value $\mathbf{Y}$:

$$\mathcal{L}_{\text{pred}} = \text{MAE}(\hat{\mathbf{Y}}, \mathbf{Y}). \tag{14}$$

At the same time, to improve the modelling consistency across stages, we introduce a stage regularisation term $\mathcal{L}_{\text{stage}}$ to constrain the intermediate results $\mathbf{X}_i$ reconstructed in $K$ stages and the original input $\mathbf{X}$:

$$\mathcal{L}_{\text{struct}} = \sum_{i=1}^{K} \text{MAE}(\mathbf{X}_i, \mathbf{X}). \tag{15}$$

The final total loss is defined as the weighted sum of the two:

$$\mathcal{L} = \mathcal{L}_{\text{pred}} + \alpha \cdot \mathcal{L}_{\text{struct}}, \tag{16}$$

where $\alpha$ controls the weight of the regularisation term.

## 4 EXPERIMENTS

### 4.1 EXPERIMENTAL SETTINGS

We evaluate PDUNet on eight widely used multivariate time series benchmarks, including ETTh1/2, ETTm1/2, Electricity, Solar-Energy, Weather, and Exchange, which cover diverse temporal patterns and forecasting challenges. Comparisons are made against ten representative models spanning different paradigms: DUET Qiu et al. (2025) and AMD Hu et al. (2025) employ decoupled modeling along temporal and variable dimensions, TimesNet Wu et al. (2023) captures periodic structures in the frequency domain, while TimeMixer Wang et al. (2024), iTransformer Liu et al. (2023a), PatchTST Nie et al. (2023), Crossformer Zhang & Yan (2023), TiDE Das et al. (2023), and DLinear Zeng et al. (2023) represent Transformer-, MLP-, or token-mixing based approaches. All models are trained with a sliding window of input length 96 and prediction lengths of 96, 192, 336, and 720, and evaluated using MSE and MAE. Experiments are implemented in Python 3.10 and run on NVIDIA RTX 4090 GPUs for efficiency and reproducibility.

### 4.2 MAIN RESULTS

The results of the main experiment are shown in Table 1, which reports the average prediction error of each method on the eight standard datasets, using both MSE and MAE metrics to assess performance. The best results are highlighted in bold and the second best results are underlined. We make the following observations from this: PDUNet achieved the best results for both MSE and MAE in six out of eight datasets, demonstrating high consistency. In particular, it significantly outperformed advanced methods such as DUET, TimeMixer, and PatchTST in medium- to long-term sequences such as ETTh1, ETTh2, ETTm1, and Weather, validating its ability to model long-term dynamics.In high-dimensional collaborative scenarios such as Solar and Electricity, PDUNet also performs robustly, significantly reducing errors while effectively preserving key structural features. Additionally, on non-stationary financial data such as Exchange, PDUNet achieves leading performance, demonstrating strong generalisation and dynamic adaptation capabilities. Overall, PDUNet maintains excellent performance across various tasks and data types, highlighting its advantages in modelling complex temporal structures.

Table 1: Multivariate time series forecasting results (averaged over all horizons). The best performance is highlighted in **red**, and the second-best is blue. Full results are in Appendix F.

| Models | PDUNet (ours) | | DUET (2025) | | AMD (2025) | | TimeMixer (2024) | | iTransformer (2024) | | PatchTST (2023) | | TimesNet (2023) | | DLinear (2023) | | Crossformer (2023) | | TiDE (2023) | |
|---|---|---|---|---|---|---|---|---|---|---|---|---|---|---|---|---|---|---|---|---|
| Metric | MSE | MAE | MSE | MAE | MSE | MAE | MSE | MAE | MSE | MAE | MSE | MAE | MSE | MAE | MSE | MAE | MSE | MAE | MSE | MAE |
| ETTh1 | **0.422** | **0.433** | 0.443 | 0.436 | 0.439 | 0.437 | 0.447 | 0.440 | 0.454 | 0.447 | 0.453 | 0.445 | 0.458 | 0.450 | 0.461 | 0.457 | 0.529 | 0.522 | 0.541 | 0.507 |
| ETTh2 | **0.347** | **0.379** | 0.372 | 0.395 | 0.378 | 0.400 | 0.364 | 0.395 | 0.383 | 0.407 | 0.366 | 0.395 | 0.414 | 0.427 | 0.563 | 0.519 | 0.942 | 0.684 | 0.611 | 0.550 |
| ETTm1 | **0.377** | **0.388** | 0.390 | 0.393 | 0.389 | 0.395 | 0.381 | 0.384 | 0.407 | 0.410 | 0.384 | 0.396 | 0.400 | 0.406 | 0.404 | 0.408 | 0.513 | 0.495 | 0.419 | 0.419 |
| ETTm2 | **0.273** | **0.318** | 0.280 | 0.324 | 0.282 | 0.327 | 0.275 | 0.323 | 0.288 | 0.332 | 0.285 | 0.328 | 0.291 | 0.333 | 0.354 | 0.402 | 0.757 | 0.610 | 0.358 | 0.404 |
| Weather | **0.235** | **0.264** | 0.251 | 0.273 | 0.261 | 0.286 | 0.240 | 0.271 | 0.258 | 0.278 | 0.258 | 0.281 | 0.259 | 0.287 | 0.265 | 0.315 | 0.264 | 0.320 | 0.271 | 0.320 |
| Electricity | 0.175 | 0.270 | **0.172** | **0.258** | 0.184 | 0.271 | 0.182 | 0.279 | 0.178 | 0.270 | 0.203 | 0.284 | 0.192 | 0.304 | 0.225 | 0.319 | 0.244 | 0.334 | 0.251 | 0.344 |
| Solar | **0.197** | **0.247** | 0.237 | 0.233 | 0.233 | 0.250 | 0.216 | 0.280 | 0.233 | 0.262 | 0.287 | 0.333 | 0.403 | 0.374 | 0.330 | 0.401 | 0.406 | 0.442 | 0.347 | 0.417 |
| Exchange | **0.309** | **0.375** | 0.318 | 0.384 | 0.367 | 0.407 | 0.391 | 0.453 | 0.360 | 0.403 | 0.372 | 0.409 | 0.416 | 0.443 | 0.354 | 0.414 | 0.940 | 0.707 | 0.370 | 0.413 |

## 4.3 ABLATION STUDY

To evaluate the effectiveness of key components in PDUNet, we conduct three sets of ablation studies on representative datasets: ETTh2, ETTm1, and Weather. All results are reported as average MSE and MAE across four prediction lengths.

- Series A: Predictive Path Ablation. We remove the Gradient Descent Module (w/o GDM), the Proximal Mapping Module (w/o PMM), and restrict the model to single-step prediction to assess the role of iterative refinement.

- Series B: Dual-Path Structure Ablation. We replace Dual-SSD with a shared encoder and individually remove the Channel SSD or Temporal SSD to evaluate the importance of separate modelling along channel and temporal dimensions.

- Series C: State Mixing Strategy Ablation. We test two variants with fixed or causal forms of the mixing matrix $A$, to study how different state composition strategies affect model expressiveness.

Table 2: Ablation results on ETTh2, ETTm1, and Weather. We report the average MSE and MAE over four prediction lengths.

| Model | ETTh2 | | ETTm1 | | Weather | |
|---|---|---|---|---|---|---|
| Metrics | MSE | MAE | MSE | MAE | MSE | MAE |
| PDUNet (Full) | **0.347** | **0.379** | **0.377** | **0.388** | **0.235** | **0.264** |
| *A Series – Forecasting Optimization Modules* | | | | | | |
| w/o GDM | 0.364 | 0.395 | 0.392 | 0.403 | 0.247 | 0.278 |
| w/o PMM | 0.358 | 0.391 | 0.388 | 0.398 | 0.243 | 0.273 |
| One-step only | 0.355 | 0.387 | 0.385 | 0.394 | 0.240 | 0.271 |
| *B Series – Dual-SSD Feature Modelling* | | | | | | |
| Replace Dual-SSD | 0.359 | 0.390 | 0.390 | 0.398 | 0.248 | 0.279 |
| w/o Channel SSD | 0.362 | 0.392 | 0.394 | 0.401 | 0.250 | 0.280 |
| w/o Temporal SSD | 0.361 | 0.390 | 0.393 | 0.400 | 0.247 | 0.278 |
| *C Series – State Mixing Strategies* | | | | | | |
| Fixed A Matrix | 0.356 | 0.387 | 0.385 | 0.395 | 0.242 | 0.271 |
| Causal A Matrix | 0.353 | 0.386 | 0.383 | 0.393 | 0.239 | 0.270 |

The experimental results are shown in Table 2, and we have the following key observations:

1) The iterative modelling mechanism is essential. Removing either the gradient descent or proximal mapping module leads to notable performance drops, confirming the importance of multi-step updates.

2) The dual-path structure for channel and temporal modelling is critical. Replacing it with a shared encoder or removing one path significantly harms performance, supporting the effectiveness of decoupled modelling.

3) State mixing strategies impact model expressiveness. Fixed or causal matrices perform worse, while the flexible non-causal approach in CHSM-SSD better adapts to dynamic structures.

## 4.4 MODEL ANALYSIS

### 4.4.1 HYPERPARAMETER SENSITIVITY.

We performed sensitivity analyses on three key hyperparameters, namely the proximal regular term weights $\alpha$, the number of iteration rounds $K$ and the patch length $P$, and the experimental results are shown in Table 3. We conduct experiments on three typical datasets (ETTh2, ETTm1 and Weather) to analyse their impact on the model prediction performance (MSE and MAE). Based on the analyses, we obtain the following key observations:

1) Moderate regularisation weights (e.g., $\alpha = 0.1$) help stabilise optimisation, while excessively large weights suppress model learning.

2) Increasing the number of iterations $K$ can improve performance, but too many iterations yield limited benefits and may lead to overfitting. We ultimately set $K = 5$.

3) The patch scale $P$ significantly affects the model's modelling capability. A smaller $P$ (e.g., 8) performs best on multi-dataset scenarios, avoiding information compression losse

In addition, the step size $\eta$ is a learnable parameter, so it is not involved in the hyperparametric analysis.

Table 3: Hyperparameter sensitivity analysis on ETTh1, ETTm1 and Weather.

| Dataset | $\alpha$ | | | | $K$ | | | | $P$ | | |
|---------|------|------|-------|-------|-------|-------|-------|-------|-------|-------|-------|
| Metrics | 0.01 | 0.1 | 1 | 10 | 3 | 5 | 8 | 10 | 8 | 16 | 32 |
| ETTh1 | 0.441 | **0.422** | 0.432 | 0.453 | 0.439 | **0.422** | 0.425 | 0.429 | **0.422** | 0.429 | 0.436 |
| ETTm1 | 0.389 | **0.377** | 0.381 | 0.392 | 0.388 | **0.377** | 0.380 | 0.384 | **0.377** | 0.382 | 0.386 |
| Weather | 0.248 | **0.235** | 0.241 | 0.249 | 0.243 | **0.235** | 0.239 | 0.244 | **0.235** | 0.239 | 0.244 |

### 4.4.2 VARYING LOOK-BACK WINDOW.

As shown in Table 4, PDUNet demonstrates good adaptability and robustness under different look-back windows. Overall, it shows a trend of 'the longer the history, the lower the error,' which is particularly significant in long prediction tasks (such as $F = 720$). Even in medium and short windows (such as $H = 192$), the model maintains stable performance, demonstrating its ability to efficiently model historical information.

Table 4: Performance comparison of PDUNet with different look-back window lengths ($H$) on ETTh1 and Weather datasets.

| Dataset | $H$=48 | | $H$=96 | | $H$=192 | | $H$=336 | | $H$=512 | | $H$=720 | |
|---------|------|------|------|------|------|------|------|------|------|------|------|------|
| Metrics | MSE | MAE | MSE | MAE | MSE | MAE | MSE | MAE | MSE | MAE | MSE | MAE |
| ETTh1 ($R$=96) | 0.373 | 0.398 | 0.367 | 0.392 | 0.359 | 0.389 | 0.354 | 0.388 | 0.351 | 0.388 | **0.348** | **0.387** |
| ETTh1 ($R$=720) | 0.486 | 0.494 | 0.475 | 0.482 | 0.462 | 0.471 | 0.445 | 0.463 | 0.435 | 0.455 | **0.430** | **0.451** |
| Weather ($R$=96) | 0.157 | 0.196 | 0.151 | 0.191 | 0.149 | 0.189 | 0.148 | 0.189 | 0.148 | 0.190 | **0.147** | **0.189** |
| Weather ($R$=720) | 0.331 | 0.339 | 0.326 | 0.334 | 0.318 | 0.328 | 0.314 | 0.324 | 0.310 | 0.321 | **0.307** | **0.319** |

## 5 CONCLUSION

In this paper, we present a novel prediction framework, PDUNet, which tightly integrates model-driven and data-driven paradigms through a learnable proximal unrolling process. Specifically, PDUNet formulates the prediction task as an iterative optimisation problem and introduces a dual structural state decoder (Dual-SSD) to capture structural dependencies in both time and channel dimensions. The whole process is modelled as an interpretable and controllable evolutionary path guided by gradient descent and approximate mapping. This design allows PDUNet to co-optimise prediction accuracy and structural consistency during the iterative process. Extensive experiments on eight benchmark datasets show that PDUNet outperforms state-of-the-art methods, especially for long-horizon prediction.

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

## A  APPENDIX

## B  RELATED WORK

### B.1  TRADITIONAL STATISTICAL MODELS

Before the widespread adoption of deep learning, time series forecasting primarily relied on traditional statistical models, which laid the foundation for sequential data analysis. Exponential smoothing applies exponentially decaying weights to historical observations, emphasizing more recent data. It demonstrates high computational efficiency and adaptability in real-time scenarios Brown (1959); Holt (1957); Winters (1960). Another representative approach is the Autoregressive Integrated Moving Average (ARIMA) model, which combines autoregressive (AR), moving average (MA), and differencing (I) components to effectively capture linear dependencies in time series data Box et al. (2015). The AR component uses linear combinations of past values, the MA component adjusts for past errors, and the I component removes non-stationarity through differencing. ARIMA is suitable for medium- to long-term forecasting, and its seasonal extension, SARIMA, incorporates seasonal differencing and periodic structures, enabling the modeling of time series with cyclical patterns. In addition, the Prophet model has gained popularity in business time series forecasting due to its usability and interpretability Navratil & Kolkova (2019). It adopts an additive decomposition framework that explicitly models trend, seasonality, and holiday effects, offering strong scalability and robustness for data with significant seasonal structure or abrupt changes.

Despite their theoretical transparency and interpretability, these statistical models are largely based on linear assumptions and lack the capacity to model high-dimensional interactions or complex non-stationary dynamics, limiting their applicability in modern multivariate forecasting tasks.

### B.2  DEEP LEARNING METHODS

With the advancement of hardware capabilities and the explosive growth of data, deep learning-based methods have been widely applied in time series forecasting across various domains. Existing research can be broadly categorized into two strategies: Channel-Independent (CI) and Channel-Dependent (CD).

The channel-independent strategy assumes that different channels are mutually independent and thus shares the same model weights across all channels while modeling each one separately. This approach is structurally simple and computationally efficient, significantly reducing training overhead. For example, DLinear Zeng et al. (2023) employs a highly simplified linear model and yet outperforms many Transformer-based approaches. PatchTST Nie et al. (2023) utilizes temporal slicing and Transformer-based feature extraction to effectively capture single-channel temporal dependencies. Additionally, SparseTSF Lin et al. (2024) decouples seasonal and trend components for separate modeling, achieving competitive results under the CI framework. In contrast, the channel-dependent strategy explicitly models inter-channel correlations, aiming to enhance prediction capacity through cross-variable interaction. This is typically achieved through shared information, joint modeling, or structure-guided learning. Representative models include CSformer Wang et al., which introduces dimension-wise embedding and a two-stage attention mechanism to effectively capture hidden dependencies in multivariate time series; Ada-MSHyper Shang et al. (2024), which employs graph networks to support group-wise interactions; and DUET Qiu et al. (2025), which integrates clustering to dynamically model complex inter-channel relationships and improve channel-specific prediction accuracy.

While traditional statistical models offer strong interpretability during modeling, deep learning methods demonstrate notable advantages in representation capacity and predictive performance. However, mainstream approaches still face a critical challenge: the lack of effective integration

of structural priors into deep models, which limits their adaptability to non-stationarity, structural shifts, and complex interdependencies. To address this issue, we propose a Proximal Deep Unrolling (PDU) network, which integrates structural modeling with optimization-driven mechanisms. By introducing learnable proximal operators and a state-driven structural modulation mechanism, the proposed method dynamically perceives temporal structures and variable relationships, enabling accurate and robust modeling of non-stationary, multi-scale multivariate time series.

## C   ALGORITHM

We provide an algorithm flow 1 for the proximal unrolling method.

---

**Algorithm 1** Our Proximal Forecasting Network

---

**Require:** Time series $X$, structure matrix $E$, step size $\eta$, regularisation weight $\lambda$, number of iterations $K$
**Ensure:** Predicted future sequence $\hat{Y}^{(K)}$
1: $Y \leftarrow XE$                                    {Extract history via structural projection}
2: Initialise $\hat{Y}^{(0)} \leftarrow \mathbf{0}$
3: **for** $k = 0$ to $K - 1$ **do**
4:     $G^{(k)} \leftarrow (YA + \hat{Y}^{(k)}B)E - Y$                     {Reconstruction error}
5:     $\nabla_{\hat{Y}} g^{(k)} \leftarrow G^{(k)}(BE)^{\top}$                     {Gradient computation}
6:     $\hat{Y}^{(k+1)} \leftarrow prox_{\eta,\lambda\mathcal{J}}\left(\hat{Y}^{(k)} - \eta\nabla_{\hat{Y}} g^{(k)}\right)$                     {Proximal update}
7: **end for**
8: **return** $\hat{Y}^{(K)}$

---

## D   BACKGROUND ON SSD AND NC-SSD

### D.1   STATE SPACE DUALITY (SSD)

We adopt the state space duality (SSD) framework introduced in Mamba2 Dao & Gu (2024), which reformulates the sequence modeling process as a structured state update. Let $\mathbf{x} \in \mathbb{R}^{L \times D}$ be the token sequence, the SSD transformation is defined as:

$$\mathbf{y} = \mathrm{SSD}(\mathbf{x}, a, \mathbf{B}, \mathbf{C}) = \left(\mathbf{M} \cdot (\mathbf{C}\mathbf{B}^{\top})\right)\mathbf{x},$$

where $\mathbf{B}, \mathbf{C} \in \mathbb{R}^{L \times N}$ are the input and output projections to state space, and $a \in \mathbb{R}^L$ controls the decay strength of each token. The matrix $\mathbf{M} \in \mathbb{R}^{L \times L}$ is a lower triangular control mask, which is used to construct its causality, as shown below:

$$\mathbf{M}_{ij} = \begin{cases} \prod_{k=j+1}^{i} a_k, & i > j \\ 1, & i = j \\ 0, & i < j \end{cases}.$$

### D.2   NON-CAUSAL STATE SPACE DUALITY (NC-SSD)

In order to improve parallel efficiency and adapt to the requirements of causal-free modelling, NC-SSD Lee et al. further simplifies the construction of the control matrix $\mathbf{M}$ based on the standard SSD. Its core idea is to eliminate the modelling constraints caused by causal masking by uniformly weighting and aggregating each label using a unified decay factor $a$. Its specific form is as follows:

$$\mathbf{y} = \mathrm{NCSSD}(\mathbf{x}, a, \mathbf{B}, \mathbf{C}) = \mathbf{C}\mathbf{h},$$

where the hidden state $\mathbf{h}$ is denoted as:

$$\mathbf{h} = \left(a \cdot \mathbf{1}_N^{\top} \odot \mathbf{B}\right)^{\top} \mathbf{x} = \sum_{i=1}^{L} a_i \mathbf{B}_i^{\top} \mathbf{x}_i,$$

Here, $\mathbf{1}_N \in \mathbb{R}^N$ is the broadcast vector, $\mathbf{B}_i^\top \mathbf{x}_i$ denotes the feature representation of each token after mapping it to the state space, and $a_i$ provides the global weight of each position.

All computations for standard SSD and NC-SSD are performed in token space, so the overall computational complexity is the same as that of standard SSD, i.e.

$$\mathcal{O}(LD^2 + LND).$$

## E  DATASET DETAILS

We evaluate our method on eight widely-used multivariate time series forecasting datasets. A brief description of each dataset is provided below:

- **ETTh1 / ETTh2**: Hourly power transformer data from the ETT benchmark, including oil temperature, ambient temperature, and other load indicators. ETTh1 and ETTh2 represent different transformer devices.
- **ETTm1 / ETTm2**: Similar to ETTh datasets but sampled at 15-minute intervals. ETTm1 and ETTm2 also correspond to different transformer instances.
- **Electricity**: Hourly electricity consumption records from 321 customers. Widely used for testing models' capability on high-dimensional and periodic data.
- **Solar-Energy**: 10-minute solar power output data collected from 137 photovoltaic plants in the United States. Represents high-dimensional data with strong periodic patterns.
- **Weather**: Meteorological data collected every 10 minutes in 2020, including 21 environmental indicators such as temperature, humidity, wind speed, and pressure.
- **Exchange**: Daily exchange rates between eight foreign currencies, reflecting non-stationary economic dynamics and challenging long-horizon prediction scenarios.

## F  FULL FORECASTING RESULTS

The full multivariate forecasting results are provided in the following section due to the space limitation of the main text. Table 5 presents the detailed multivariate results of all prediction lengths in terms of MSE/MAE across eight well.acknowledged benchmarks.

## G  ROBUSTNESS ANALYSIS

To assess the robustness of PDUNet against input perturbations, we inject Gaussian noise into the test input, defined as:

$$X' = X + \epsilon, \quad \epsilon \sim \mathcal{N}(0, \sigma^2).$$

We evaluate the model under different prediction lengths (96, 192, 336, 720) on the ETTh1 dataset, and report the corresponding MSE and MAE. As shown in Figure 5, the MSE remains relatively stable as noise increases, with only slight variations across different settings. These non-monotonic fluctuations reflect natural instability caused by input noise and optimization randomness, rather than structural sensitivity, indicating the inherent robustness of the model.

### G.1  EFFICIENCY ANALYSIS

We evaluate the parameter efficiency of PDUNet on the ETTh1 dataset and compare it with several representative baselines under varying future and history sequence lengths. The results in Table 6 show that PDUNet maintains a relatively compact architecture across all settings. Compared to mainstream models, it achieves a favorable balance between modeling capacity and parameter cost, making it well-suited for scalable and lightweight forecasting scenarios.

Table 5: Main forecasting results across 8 datasets. Best in **bold**, second best underlined.

| Models | | PDUNet (ours) | | DUET (2025) | | AMD (2025) | | TimeMixer (2024) | | iTransformer (2024) | | PatchTST (2023) | | TimesNet (2023) | | DLinear (2023) | | Crossformer (2023) | | TiDE (2023) | |
|---|---|---|---|---|---|---|---|---|---|---|---|---|---|---|---|---|---|---|---|---|---|
| Metric | | MSE | MAE | MSE | MAE | MSE | MAE | MSE | MAE | MSE | MAE | MSE | MAE | MSE | MAE | MSE | MAE | MSE | MAE | MSE | MAE |
| ETTh1 | 96 | **0.367** | **0.392** | 0.377 | 0.393 | 0.379 | 0.394 | 0.375 | 0.400 | 0.386 | 0.405 | 0.393 | 0.408 | 0.384 | 0.402 | 0.397 | 0.412 | 0.423 | 0.448 | 0.479 | 0.464 |
| | 192 | **0.419** | 0.422 | 0.429 | 0.425 | 0.433 | 0.423 | 0.429 | **0.421** | 0.441 | 0.512 | 0.445 | 0.434 | 0.436 | 0.429 | 0.446 | 0.441 | 0.471 | 0.474 | 0.525 | 0.492 |
| | 336 | **0.426** | **0.434** | 0.471 | 0.446 | 0.465 | 0.446 | 0.484 | 0.458 | 0.487 | 0.458 | 0.484 | 0.451 | 0.491 | 0.469 | 0.489 | 0.467 | 0.570 | 0.546 | 0.565 | 0.515 |
| | 720 | **0.475** | 0.482 | 0.496 | **0.480** | 0.479 | 0.483 | 0.498 | 0.482 | 0.503 | 0.491 | 0.491 | 0.485 | 0.521 | 0.500 | 0.513 | 0.510 | 0.653 | 0.621 | 0.594 | 0.558 |
| | Average | **0.422** | **0.433** | 0.443 | 0.436 | 0.439 | 0.437 | 0.447 | 0.440 | 0.454 | 0.447 | 0.453 | 0.445 | 0.458 | 0.450 | 0.461 | 0.457 | 0.529 | 0.522 | 0.541 | 0.507 |
| ETTh2 | 96 | **0.280** | 0.332 | 0.296 | 0.337 | 0.285 | 0.341 | 0.289 | 0.341 | 0.297 | 0.349 | 0.294 | 0.343 | 0.340 | 0.374 | 0.340 | 0.394 | 0.745 | 0.584 | 0.400 | 0.440 |
| | 192 | **0.349** | **0.377** | 0.368 | 0.389 | 0.387 | 0.394 | 0.372 | 0.399 | 0.380 | 0.400 | 0.377 | 0.393 | 0.402 | 0.414 | 0.482 | 0.479 | 0.877 | 0.656 | 0.528 | 0.509 |
| | 336 | **0.354** | **0.381** | 0.411 | 0.422 | 0.416 | 0.414 | 0.386 | 0.414 | 0.428 | 0.432 | 0.381 | 0.409 | 0.452 | 0.452 | 0.591 | 0.541 | 1.043 | 0.731 | 0.643 | 0.571 |
| | 720 | **0.404** | **0.428** | 0.412 | 0.434 | 0.422 | 0.433 | 0.412 | 0.434 | 0.427 | 0.445 | 0.412 | 0.433 | 0.462 | 0.468 | 0.839 | 0.661 | 1.104 | 0.763 | 0.874 | 0.679 |
| | Average | **0.347** | **0.379** | 0.372 | 0.395 | 0.378 | 0.400 | 0.364 | 0.395 | 0.383 | 0.407 | 0.366 | 0.395 | 0.414 | 0.427 | 0.563 | 0.519 | 0.942 | 0.684 | 0.611 | 0.550 |
| ETTm1 | 96 | **0.316** | **0.347** | 0.324 | 0.354 | 0.322 | 0.359 | 0.320 | 0.357 | 0.334 | 0.368 | 0.321 | 0.360 | 0.338 | 0.375 | 0.346 | 0.374 | 0.404 | 0.426 | 0.364 | 0.387 |
| | 192 | **0.361** | **0.372** | 0.369 | 0.379 | 0.367 | 0.381 | **0.361** | 0.381 | 0.390 | 0.393 | 0.362 | 0.384 | 0.374 | 0.387 | 0.382 | 0.391 | 0.450 | 0.451 | 0.398 | 0.404 |
| | 336 | **0.385** | **0.397** | 0.404 | 0.402 | 0.400 | 0.402 | 0.390 | 0.404 | 0.426 | 0.420 | 0.392 | 0.402 | 0.410 | 0.411 | 0.415 | 0.415 | 0.532 | 0.515 | 0.428 | 0.425 |
| | 720 | **0.446** | **0.434** | 0.463 | 0.437 | 0.465 | 0.437 | 0.454 | 0.441 | 0.491 | 0.459 | 0.461 | 0.439 | 0.478 | 0.450 | 0.473 | 0.451 | 0.666 | 0.589 | 0.487 | 0.461 |
| | Average | **0.377** | **0.388** | 0.390 | 0.393 | 0.389 | 0.395 | 0.381 | 0.384 | 0.407 | 0.410 | 0.384 | 0.396 | 0.400 | 0.406 | 0.404 | 0.408 | 0.513 | 0.495 | 0.419 | 0.419 |
| ETTm2 | 96 | **0.171** | **0.252** | 0.174 | 0.255 | 0.180 | 0.265 | 0.175 | 0.255 | 0.180 | 0.264 | 0.178 | 0.260 | 0.187 | 0.267 | 0.193 | 0.293 | 0.287 | 0.366 | 0.207 | 0.305 |
| | 192 | **0.235** | **0.295** | 0.243 | 0.302 | 0.244 | 0.304 | 0.237 | 0.299 | 0.250 | 0.309 | 0.249 | 0.307 | 0.249 | 0.309 | 0.284 | 0.361 | 0.414 | 0.492 | 0.290 | 0.364 |
| | 336 | **0.295** | **0.333** | 0.304 | 0.341 | 0.302 | 0.340 | 0.298 | 0.340 | 0.311 | 0.348 | 0.321 | 0.346 | 0.321 | 0.351 | 0.382 | 0.429 | 0.597 | 0.542 | 0.377 | 0.422 |
| | 720 | **0.390** | **0.395** | 0.399 | 0.397 | 0.403 | 0.397 | 0.391 | 0.396 | 0.412 | 0.407 | 0.400 | 0.398 | 0.408 | 0.403 | 0.558 | 0.525 | 1.730 | 1.042 | 0.558 | 0.524 |
| | Average | **0.273** | **0.318** | 0.280 | 0.324 | 0.282 | 0.327 | 0.275 | 0.323 | 0.288 | 0.332 | 0.285 | 0.328 | 0.291 | 0.333 | 0.354 | 0.402 | 0.757 | 0.610 | 0.358 | 0.404 |
| Weather | 96 | **0.151** | **0.191** | 0.163 | 0.202 | 0.181 | 0.227 | 0.163 | 0.209 | 0.174 | 0.214 | 0.178 | 0.219 | 0.172 | 0.220 | 0.195 | 0.252 | 0.195 | 0.271 | 0.202 | 0.261 |
| | 192 | **0.203** | **0.242** | 0.218 | 0.252 | 0.227 | 0.264 | 0.208 | 0.250 | 0.221 | 0.254 | 0.224 | 0.259 | 0.219 | 0.261 | 0.237 | 0.295 | 0.209 | 0.277 | 0.242 | 0.298 |
| | 336 | 0.259 | **0.288** | 0.274 | 0.294 | 0.281 | 0.302 | **0.251** | **0.287** | 0.278 | 0.296 | 0.278 | 0.298 | 0.280 | 0.306 | 0.282 | 0.331 | 0.273 | 0.332 | 0.287 | 0.335 |
| | 720 | **0.326** | **0.334** | 0.349 | 0.343 | 0.356 | 0.349 | 0.339 | 0.341 | 0.358 | 0.347 | 0.353 | 0.346 | 0.365 | 0.359 | 0.345 | 0.382 | 0.379 | 0.401 | 0.351 | 0.386 |
| | Average | **0.235** | **0.264** | 0.251 | 0.273 | 0.261 | 0.286 | 0.240 | 0.271 | 0.258 | 0.278 | 0.258 | 0.281 | 0.259 | 0.287 | 0.265 | 0.315 | 0.264 | 0.320 | 0.271 | 0.320 |
| Electricity | 96 | 0.151 | 0.247 | **0.145** | **0.233** | 0.168 | 0.239 | 0.153 | 0.247 | 0.148 | 0.240 | 0.180 | 0.259 | 0.168 | 0.272 | 0.210 | 0.302 | 0.219 | 0.314 | 0.237 | 0.329 |
| | 192 | 0.162 | 0.256 | 0.163 | **0.248** | 0.169 | 0.258 | 0.166 | 0.256 | **0.162** | 0.253 | 0.188 | 0.268 | 0.184 | 0.322 | 0.210 | 0.305 | 0.231 | 0.322 | 0.236 | 0.330 |
| | 336 | 0.177 | 0.277 | **0.175** | **0.262** | 0.179 | 0.271 | 0.185 | 0.277 | 0.178 | 0.269 | 0.203 | 0.288 | 0.198 | 0.300 | 0.223 | 0.319 | 0.246 | 0.337 | 0.249 | 0.344 |
| | 720 | 0.205 | 0.301 | **0.204** | **0.291** | 0.221 | 0.314 | 0.225 | 0.310 | 0.225 | 0.317 | 0.239 | 0.321 | 0.220 | 0.320 | 0.258 | 0.350 | 0.280 | 0.363 | 0.284 | 0.373 |
| | Average | 0.175 | 0.270 | **0.172** | **0.258** | 0.184 | 0.271 | 0.182 | 0.279 | 0.178 | 0.270 | 0.203 | 0.284 | 0.192 | 0.304 | 0.225 | 0.319 | 0.244 | 0.334 | 0.251 | 0.344 |
| Solar | 96 | **0.160** | 0.209 | 0.200 | **0.207** | 0.204 | 0.211 | 0.189 | 0.259 | 0.203 | 0.237 | 0.265 | 0.323 | 0.373 | 0.358 | 0.290 | 0.378 | 0.232 | 0.302 | 0.312 | 0.399 |
| | 192 | **0.190** | 0.243 | 0.228 | **0.233** | 0.219 | 0.248 | 0.222 | 0.283 | 0.233 | 0.261 | 0.288 | 0.332 | 0.397 | 0.376 | 0.320 | 0.398 | 0.371 | 0.410 | 0.339 | 0.416 |
| | 336 | **0.211** | 0.263 | 0.262 | **0.244** | 0.258 | 0.269 | 0.231 | 0.292 | 0.248 | 0.273 | 0.301 | 0.339 | 0.420 | 0.380 | 0.353 | 0.415 | 0.495 | 0.515 | 0.368 | 0.430 |
| | 720 | **0.228** | 0.270 | 0.258 | **0.249** | 0.251 | 0.272 | 0.223 | 0.285 | 0.249 | 0.275 | 0.295 | 0.336 | 0.420 | 0.381 | 0.357 | 0.413 | 0.526 | 0.542 | 0.370 | 0.425 |
| | Average | **0.197** | 0.247 | 0.237 | **0.233** | 0.233 | 0.250 | 0.216 | 0.280 | 0.233 | 0.262 | 0.287 | 0.333 | 0.403 | 0.374 | 0.330 | 0.401 | 0.406 | 0.442 | 0.347 | 0.417 |
| Exchange | 96 | **0.082** | **0.204** | 0.086 | 0.205 | 0.083 | 0.205 | 0.090 | 0.235 | 0.086 | 0.206 | 0.088 | 0.205 | 0.107 | 0.234 | 0.088 | 0.218 | 0.256 | 0.367 | 0.094 | 0.218 |
| | 192 | **0.174** | 0.301 | 0.182 | 0.305 | 0.176 | 0.303 | 0.187 | 0.343 | 0.177 | 0.299 | 0.176 | 0.299 | 0.226 | 0.344 | 0.176 | 0.315 | 0.470 | 0.509 | 0.184 | 0.307 |
| | 336 | 0.317 | 0.407 | **0.310** | **0.403** | 0.319 | 0.418 | 0.353 | 0.473 | 0.331 | 0.417 | 0.321 | 0.416 | 0.367 | 0.448 | 0.313 | 0.427 | 1.268 | 0.883 | 0.349 | 0.431 |
| | 720 | **0.660** | **0.587** | 0.693 | 0.624 | 0.891 | 0.702 | 0.934 | 0.761 | 0.847 | 0.691 | 0.901 | 0.714 | 0.964 | 0.746 | 0.839 | 0.695 | 1.767 | 1.068 | 0.852 | 0.698 |
| | Average | **0.309** | **0.375** | 0.318 | 0.384 | 0.367 | 0.407 | 0.391 | 0.453 | 0.360 | 0.403 | 0.372 | 0.409 | 0.416 | 0.443 | 0.354 | 0.414 | 0.940 | 0.707 | 0.370 | 0.413 |

Table 6: Model Parameter Efficiency Analysis (M). Evaluation is conducted on ETTh1 under different future and history sequence lengths.

| Setting | Length | Informer | Autoformer | FEDformer | PatchTST | TimesNet | PDUNet |
|---|---|---|---|---|---|---|---|
| Future Length | 96 | 11.33 | 10.54 | 16.12 | 7.49 | 37.53 | 4.53 |
| | 192 | 11.33 | 10.54 | 16.12 | 7.49 | 37.53 | 4.79 |
| | 336 | 11.33 | 10.54 | 16.12 | 7.49 | 37.53 | 6.06 |
| | 720 | 11.33 | 10.54 | 16.12 | 7.49 | 37.53 | 11.48 |
| History Length | 96 | 11.33 | 10.54 | 16.12 | 7.49 | 37.53 | 4.18 |
| | 192 | 11.33 | 10.54 | 16.12 | 7.49 | 37.53 | 4.53 |
| | 336 | 11.33 | 10.54 | 16.12 | 7.49 | 37.53 | 5.06 |
| | 720 | 11.33 | 10.54 | 16.12 | 7.49 | 37.53 | 6.47 |

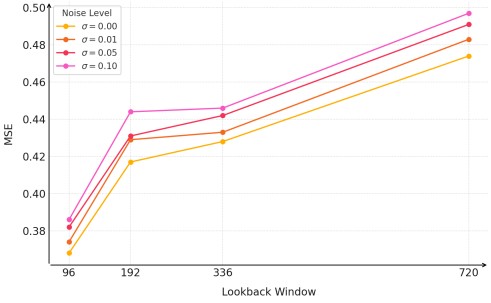

Figure 5: MSE of PDUNet under different levels of Gaussian noise on the ETTh1 dataset. Each curve represents a specific noise level $\sigma$, and a slight random fluctuation is observed due to the natural variability in model optimization and input perturbations.

