# OpenReview forum: "PDUNet: Proximal-Guided Deep Unrolling Network for Time Series Forecasting"
_ICLR.cc/2026/Conference — Submitted to ICLR 2026_

### Official Review · Reviewer_gt7t · 2025-10-19

**Soundness:** 2
**Presentation:** 3
**Contribution:** 2
**Rating:** 4
**Confidence:** 4

**Summary:**

This paper proposes PDUNet, a novel framework for time series forecasting that integrates model-driven and data-driven paradigms. The core idea is to formulate forecasting as an iterative optimization problem where the future sequence is treated as a latent variable and progressively refined. This is achieved by unrolling a proximal gradient descent algorithm into a deep network, comprising a Gradient Descent Module (GDM) and a Proximal Mapping Module (PMM). The PMM leverages a Dual-SSD structure, with separate branches (Temporal-SSD and Channel-SSD) to capture temporal and inter-channel dependencies. Extensive experiments on eight benchmark datasets demonstrate that PDUNet achieves state-of-the-art performance, particularly in long-term forecasting horizons.

**Strengths:**

1.  Moving beyond the standard "static mapping" paradigm and explicitly modeling the future as an optimizable variable within an unrolled optimization loop is a compelling and novel direction. This provides a clear, interpretable, and controllable inference path, which is often missing in deep learning-based forecasters.

2. This paper provides a rigorous formulation, from the proximal-based optimization objective (Eq. 4) to its gradient derivation and unrolling into a learnable architecture (GDM, PMM).

3. This paper is generally well-written. Figures 1 and 2 effectively illustrate the core conceptual shift and the model's architecture, making the proposed methodology easy to grasp.

**Weaknesses:**

1. This paper empirically shows that iterative updates improve performance, but it lacks any theoretical grounding. There is no discussion on the convergence guarantees of the unrolled proximal gradient method under the learned parameters.

2. This paper repeatedly emphasizes the "interpretability" and "controllability" of the proposed framework. However, no qualitative analysis or visualization is provided to demonstrate this. For instance, visualizing how the prediction $\hat{\mathbf{Y}}^{(k)}$ evolves across iterations $K$ or analyzing the learned prior $\mathcal{J}(\cdot)$ would be necessary to substantiate these claims and provide tangible insights into the model's decision-making process.

3. The selection of baselines is good but could be strengthened by including recent models like TimeBridge or TQNet that have shown strong performance in time series.

**Questions:**

Refer to Weaknesses.

---

### Official Review · Reviewer_nPrH · 2025-10-28

**Soundness:** 3
**Presentation:** 3
**Contribution:** 2
**Rating:** 4
**Confidence:** 4

**Summary:**

This paper proposes PDUNet, which formulates long-horizon multivariate time series forecasting as a structured optimization problem over future variables and solves it via proximal gradient methods combined with deep unrolling. Each iteration consists of a Gradient Descent Module (GDM) followed by a Proximal Mapping Module (PMM). The PMM contains a parallel Dual-SSD design—Temporal-SSD and Channel-SSD—that performs state-space modeling along the time and channel axes, yielding a progressively refined prediction trajectory. Training uses a combination of the primary forecasting loss and a stage-consistency regularizer. Across eight standard datasets and multiple strong baselines, PDUNet shows stable advantages for medium/long horizons, and ablations verify the necessity of the unrolling mechanism and dual-path state-space modeling.

**Strengths:**

S1. The PMM employs Dual-SSD (time/channel-parallel state-space units), shifting expensive nonlinear computation into a compressed state space, balancing expressiveness and efficiency.

S2. The algorithmic pipeline, module boundaries, and data flow are clear.

S3. Comparisons are fairly comprehensive across datasets and horizons; performance is robust for long-horizon forecasting.

**Weaknesses:**

W1. Early deep unrolling works have already networked “gradient steps + proximal mappings” (e.g., LISTA [1]); subsequent Plug-and-Play / proximal-unrolling lines further integrated learnable priors into iterative frameworks [2]. In the sequence domain, optimization- or recursion-based prediction/filtering (e.g., KalmanNet [3]) and continuous-dynamics modeling (Neural ODE [4]) are also representative. PDUNet’s contribution, explicitly optimizing future targets and realizing a structured “proximal” map via Dual-SSD, is interesting, but quantitative and mechanistic contrasts against these main lines (to clarify fundamental differences and irreplaceability) remain limited.

W2. The Dual-SSD design is methodologically close to the SSM/S4 family (Gu et al., 2021) and Mamba (Gu & Dao, 2024) for efficient long-sequence modeling; the paper also compares to iTransformer (Liu et al., 2023), PatchTST (Nie et al., 2023), DLinear (Zeng et al., 2022), and TimeMixer (Chen et al., 2024). A more systematic contrast on complexity, spectral/phase stability, and long-horizon degradation would better position PDUNet’s strengths and limitations relative to these lines.

W3. While the paper emphasizes progressive convergence, it lacks quantitative evidence such as iteration-wise error/step convergence curves and consistency of selected key lags across random seeds.
W4. The PMM effectively learns a data-driven proximal operator, yet the corresponding prior/regularizer
J(⋅) is not sufficiently interpreted or visualized; instability in J could lead to pattern leakage.

References
[1] Gregor, K., & LeCun, Y. “Learning Fast Approximations of Sparse Coding.” ICML, 2010, pp. 399–406.
[2] Venkatakrishnan, S. V., Bouman, C. A., & Wohlberg, B. “Plug-and-Play Priors for Model Based Reconstruction.” IEEE GlobalSIP, 2013, pp. 945–948.
[3] Revach, G., Shlezinger, N., Ni, X., et al. “KalmanNet: Neural Network Aided Kalman Filtering for Partially Known Dynamics.” IEEE Transactions on Signal Processing, 70:1532–1547, 2022.
[4] Chen, R. T. Q., Rubanova, Y., Bettencourt, J., & Duvenaud, D. “Neural Ordinary Differential Equations.” NeurIPS, 2018.

**Questions:**

Q1. Under channel permutation and channel-subset missing perturbations, do the factors/states of Dual-SSD remain similar?

Q2. Under the same complexity budget (aligned parameters/memory/latency), how does PDUNet compare with S4/Mamba for long-horizon forecasting performance?

---

### Official Review · Reviewer_MwUo · 2025-11-01

**Soundness:** 3
**Presentation:** 2
**Contribution:** 3
**Rating:** 4
**Confidence:** 5

**Summary:**

This paper proposes the Proximal Deep Unfolding Network (PDUNet), a coupled closed-loop prediction framework that begins with modeling, leverages data-driven mechanisms for resolution, and progressively optimizes the inference path. Unlike prior work that treats prediction targets solely as training supervision signals, the paper establishes an optimizable prediction equation based on the coupling between future variables and historical inputs, employing approximate optimization algorithms for gradual decoupling and updating. Extensive experiments demonstrate the effectiveness of PDUNet.

**Strengths:**

1. The method is relatively novel. The paper establishes an optimizable prediction equation based on the coupling between future variables and historical inputs, employing an approximate optimization algorithm for gradual decoupling and updating. Furthermore, the paper unfolds this optimization process into a dual-branch state space structure. Temporal-SSD and Channel-SSD jointly achieve progressive inference and dynamic prediction.

2. State-of-the-art performance. As shown in Table 1, the paper achieves state-of-the-art performance on most datasets.

**Weaknesses:**

1. Lack of visual experiments. While the paper performs well in quantitative experiments, it lacks complementary qualitative experiments. For example: Do the prediction curves fit the true values? Are the forecasts for cycles and trends accurate? Can extreme values be predicted? The paper needs to provide some visual experiments.

2. Lack of comparison with recent methods. I note that the paper employs SSM, so highly relevant prior work should be compared, including both methodology and experiments. This would help demonstrate the paper's innovation and effectiveness.

[1] Chimera: Effectively modeling multivariate time series with 2-dimensional state space models (NeurIPS24)
[2] TimePro: Efficient Multivariate Long-term Time Series Forecasting with Variable-and Time-Aware Hyper-state [ICML'25]

3. I have some confusion regarding the ablation experiments in Table 3. When exploring a hyperparameter such as K, does the author keep all other parameters constant? Based on my previous experience, optimal hyperparameters often vary across different datasets. However, as shown in Table 3, the three optimal parameters are identical across all three datasets. I think this seems counterintuitive.

4. Efficiency metrics must be provided to facilitate the assessment of the method's practical efficiency. Examples include the number of parameters, FLOPs, and throughput.

5. How does this method perform in short-term forecasting? The authors did not restrict the scope to long-term time series forecasting in the title. Therefore, some short-term benchmarks such as PEMs can be provided.

**Questions:**

Please refer to the weaknesses.

---

### Official Review · Reviewer_KkV7 · 2025-11-01

**Soundness:** 2
**Presentation:** 3
**Contribution:** 2
**Rating:** 4
**Confidence:** 4

**Summary:**

In most existing time series forecasting methods, the forecasting target is only used as a training supervision signal and is not explicitly incorporated into the modeling process. This makes it difficult for models to utilize future modality information for structural constraints and path guidance. This paper proposes a new forecasting framework called PDUNet, which tightly integrates the model-driven and data-driven paradigms through a learnable proximal unfolding process. Specifically, PDUNet formulates the forecasting task as an iterative optimization problem and introduces a Dual-Structural State Decoder (Dual-SSD) to capture structural dependencies in both the temporal and channel dimensions. The entire process is modeled as an interpretable and controllable evolutionary path guided by gradient descent and proximal mapping. This design enables PDUNet to jointly optimize prediction accuracy and structural consistency during the iterative process. The paper demonstrates the effectiveness of the proposed method on multiple public datasets.

**Strengths:**

This paper proposes a new forecasting framework called PDUNet, which tightly integrates the model-driven and data-driven paradigms through a learnable proximal unfolding process. Specifically, PDUNet formulates the forecasting task as an iterative optimization problem and introduces a Dual-Structural State Decoder (Dual-SSD) to capture structural dependencies in both the temporal and channel dimensions. The entire process is modeled as an interpretable and controllable evolutionary path guided by gradient descent and proximal mapping. This design enables PDUNet to jointly optimize prediction accuracy and structural consistency during the iterative process. The paper demonstrates the effectiveness of the proposed method on multiple public datasets.

**Weaknesses:**

1. The paper formulates an optimizable prediction equation based on the coupling between future variables and historical inputs and employs a proximal optimization algorithm for step-by-step decoupling and updating. However, the code implementation is confusing and seems to differ from the theoretical description in the paper. For example, many learnable parameters are initialized in the code with annotations of their physical meanings, without implementing the corresponding functions, such as not calculating the gradients for g'. In reality, it is merely the output of neural networks. The future variable Y_PROX is also initialized to all zeros, which seems to have no explicit connection with the actual future variables.

2. The paper lacks comparative experiments on time efficiency with different baselines.

3. The experiments in the paper are not solid and convincing. The classic dataset traffic is missing. Moreover, the ablation experiments were only conducted on three small-scale datasets.

4. The experimental evaluation in the paper is not very reliable. It was only assessed on a 96-window, while different methods may be suitable for different window lengths. It would be better to evaluate on multiple window lengths, similar to DUET. For example, when the window length is 512, the method in the paper may not be better than DUET.

5. Compared to most datasets, the variance differences in some results compared to the baselines are excessively large, such as in the solar dataset. It is unclear what causes the differences.

**Questions:**

The paper proposes to use an optimizable prediction equation based on the coupling between future variables and historical inputs and employs a proximal optimization algorithm for step-by-step decoupling and updating. However, in the actual code implementation, many parts use initialized learnable parameters and neural network outputs as substitutes, and it seems that there are no additional functional constraints under the theory of proximal optimization. How can we ensure that these initialized parameters conform to the theory of proximal optimization? How can we ensure that their optimization direction is towards the ideal direction described in the paper? For example, they could be optimized to match the physical meanings of the corresponding variables annotated in the code.

---

### Meta-Review · Area_Chair_wpNH · 2026-01-05

**Summary:**

The paper proposes a method for time-series forecasting based on learning to solve a multi-step unrolled optimization process. The initial reviews were lukewarm. Reviewers raised concerns about insufficient empirical validation, unclear connections to prior work (e.g., KalmanNet and Neural ODEs), and claims regarding "interpretability" and "controllability" that were not well supported. Unfortunately, the authors did not provide a rebuttal to address these issues.

After considering the paper and reviews, the AC recommends rejection.

**Reviewer Concerns:**

The authors did not provide a rebuttal. Thus, no concerns have been addressed.

**Reviewer Scores:**

As there is no rebuttal, I expect all reviewers maintaining their initial scores.

---

### Decision · Program_Chairs · 2026-01-26

Reject